# Exogenous Melatonin Promotes Glucoraphanin Biosynthesis by Mediating Glutathione in Hairy Roots of Broccoli (*Brassica oleracea* L. var. *italica* Planch)

**DOI:** 10.3390/plants13010106

**Published:** 2023-12-29

**Authors:** Jinyu Bao, Jie Yang, Xu Lu, Lei Ma, Xiaotong Shi, Shimin Lan, Yi Zhao, Jie Cao, Shaoying Ma, Sheng Li

**Affiliations:** 1State Key Laboratory of Aridland Crop Science, Gansu Agricultural University, Lanzhou 730070, China; baojy@st.gsau.edu.cn (J.B.); malei@st.gsau.edu.cn (L.M.); 2College of Horticulture, Gansu Agricultural University, Lanzhou 730070, China; luxuwq@163.com; 3College of Life Science and Technology, Gansu Agricultural University, Lanzhou 730070, China; yexuhuangwu@163.com (J.Y.); shixt@st.gsau.edu.cn (X.S.); lansm@st.gsau.edu.cn (S.L.); zhaoyi@st.gsau.edu.cn (Y.Z.); caoj@st.gsau.edu.cn (J.C.); 4Laboratory and Practice Base Management Center, Gansu Agricultural University, Lanzhou 730070, China

**Keywords:** melatonin, glutathione, sulfur transport, glucoraphanin, sulforaphane, hairy roots

## Abstract

To investigate the mechanism of melatonin (MT)-mediated glutathione (GSH) in promoting glucoraphanin (GRA) and sulforaphane (SF) synthesis, the gene expression pattern and protein content of hairy broccoli roots under MT treatment were analyzed by a combination of RNA-seq and tandem mass spectrometry tagging (TMT) techniques in this study. Kyoto Encyclopedia of Genes and Genomes (KEGG) analysis revealed that both proteins and mRNAs with the same expression trend were enriched in the “Glutathione metabolism (ko00480)” and “Proteasome (ko03050)” pathways, and most of the differentially expressed genes (DEGs) and differentially abundant proteins (DAPs) regulating the two pathways were downregulated. The results showed that endogenous GSH concentration and GR activity were increased in hairy roots after MT treatment. Exogenous GSH could promote the biosynthesis of GRA and SF, and both exogenous MT and GSH could upregulate the expression of the *GSTF11* gene related to the sulfur transport gene, thus promoting the biosynthesis of GRA. Taken together, this study provides a new perspective to explore the complex molecular mechanisms of improving GRA and SF synthesis levels by MT and GSH regulation.

## 1. Introduction

Broccoli (*Brassica oleracea* L. var. *italica* Planch) is a Brassica food species rich in several functional phytochemicals [1]. It is high in nutrient value (such as minerals, vitamins, and dietary fiber) and bioactive substances (such as glucosinolates (GLS) and phenolic compounds), the content of which varies depending on the different tissues [2]. GLS is a secondary metabolite mainly found in cruciferous plants with a wide variety of species that contain nitrogen and sulfur in their structures [3], with a sulfur content of approximately 30% [4,5]. Sulfur is an essential nutrient for plants, and sulfur deficiency can affect plant growth and development [6]. Research has shown that sulfur deficiency may result in the loss of GLS [7]. The glucoraphanin (GRA) concerned in this study belongs to a kind of aliphatic GLS, and its downstream product, sulforaphane (SF), has received extensive attention for its anticancer effect. Plant damage activates the “GLS-Myrosinase” binary enzyme system: GRA is released from the plant cell vacuole and undergoes hydrolysis with myrosinase (MYR) present in MYR cells [8] to generate SF. SF is considered one of the most potent natural anticancer compounds for its anticancer [9], antibacterial [10], anti-inflammatory [11], and antioxidant [12] activities. Currently, the production of SF from broccoli seeds and nutrient organs is costly and low yielding, while the hairy root culture technique can effectively solve the problem of low SF yield [13,14,15,16].

Hairy roots are formed by plant wounds infested by *Agrobacterium rhizogenes*, a Gram-negative soil bacterium containing a root-inducing (Ri) plasmid [17], which carries a segment of T-DNA containing a set of genes that encode plant hormone synthases such as auxin [18]. *Agrobacterium rhizogenes* invades plant cells from plant wounds to trigger a new balance of hormones, and induces the formation of adventitious roots, called hairy roots [19]. Hairy roots are characterized by faster proliferation, hormone-free growth, and genetic stability [20]. Hairy roots have been reported to be a good source of metabolites, such as indigotin production in *Isatis tinctoria* [21], solasodine production in *Solanum trilobatum* [22], and tanshinone production in *Salvia miltiorrhiza* [23]. Kim et al. (2013) found that the use of the hairy broccoli root culture system could promptly produce GRA [24]. Previous studies in our research group showed that the addition of exogenous melatonin (MT) promoted an increase in the yield of GRA and SF in liquid medium cultured hairy broccoli roots [15,16].

MT, *N*-acetyl-5-methoxytryptamine, is a class of small molecules with pleiotropic physiological activity, involved in the regulation of plant growth and development in chemical messengers. Its discovery in vertebrates for the first time in 1958 attracted great attention [25]. Scholars had identified the structure and function of MT molecules and found that MT can regulate animal growth and development as well as participate in cellular signal transduction. Since MT was first found in animals and not found in plants for a period of time, MT was once considered to be a unique hormone in animals. It was not until 1995 that Dubbels et al. (1995) and Hattori et al. (1995), respectively, found the existence of MT in higher plants [26,27]. As an important endogenous multifunctional molecule, MT exhibits broad-spectrum antioxidant activity and plays a key role in responding to biotic and abiotic stress, regulating plant growth and development, circadian rhythm, reproductive physiology, and senescence [28,29,30]. Previous studies in our laboratory have shown that the addition of exogenous MT can promote the synthesis of GRA and SF [15,16], but the specific regulatory mechanism is still unclear.

Glutathione (γ-Glu-Cys-Gly) (GSH) is a multifunctional metabolite in plants and a major reservoir of non-protein-reducing sulfur with an important role in cellular defense and protection. GSH is the main intracellular sulfhydryl-containing compound that, in addition to direct scavenging of free radicals, acts as a cofactor for antioxidant enzymes such as superoxide dismutase (SOD), catalase (CAT), and peroxidase (POD), and regenerates the active form of antioxidants such as ascorbic acid, keeping protein sulfhydryl groups in a reduced state [31]. In the process of GSL synthesis, glutathione S-transferases (GSTs) catalyze the binding of sulfur donor and acyl nitrogen compounds to form S-alkyl-thiol esters [32,33]. However, using MT-mediated GSH to regulate GRA and SF biosynthesis has not been reported.

In this study, RNA-Seq and Tandem mass spectrometry tagging (TMT) were used to investigate the molecular mechanisms involved in the effect of exogenous MT on the transcriptional–translational level of GRA in hairy broccoli roots. In addition, endogenous GSH, glutathione reductase (GR) activity, GRA and SF production, as well as the expression of genes related to GRA and SF synthesis and sulfur transport in hairy roots under the treatment of exogenous GSH and MT have detected. This study provides a new perspective to explore the complex molecular mechanism of MT-mediated GSH transport to regulate GRA and SF synthesis.

## 2. Results

### 2.1. Integrative Analysis of the Transcriptome and Proteome 

#### 2.1.1. Correlation Analysis of the Transcriptome and Proteome

This study was performed to analyze the association between genes and proteins of hairy broccoli roots after MT treatment. To show the correlation between genes from the transcriptome and proteins from the proteome, a scatter plot was created where each point represented the fold of difference between the gene and the protein (log2FC). Since the abundance of proteins in cells was mainly controlled by the level of translation, there were significant differences between proteins and mRNAs in terms of half-life, synthesis rate, and quantity. The scatter plot was divided into nine quadrants by *x* and *y* axes, with the line on the *x*-axis indicating the threshold for ploidy change at the protein level and the line on the *y*-axis indicating the threshold for ploidy change at the mRNA level. Points outside the threshold line indicated differentially expressed proteins/genes, while points inside the threshold line indicated proteins/genes that were not significantly different. In this study, a total of 5785 proteins/genes with quantitative information were distributed in nine quadrants (Figure 1). Quadrants one and nine indicated opposite protein and mRNA expression trends; quadrants three and seven indicated the same protein and mRNA expression trends; quadrants two and eight indicated no change in protein and differential mRNA expression; quadrants four and six indicated no change in mRNA and differential protein expression; and quadrant five indicated no differential expression of both protein and mRNA. In this study quadrants one to nine had 21, 44, 3, 183, 4684, 506, 15, 199, and 132 genes/proteins respectively. A total of 199 proteomics-related mRNAs were identified (Figure 1), of which 178 genes and proteins showed inconsistent expression trends at the transcriptome level and protein level, and 21 genes showed consistent expression trends at the transcriptome level and protein level. Among these 21 genes, 11 genes and their encoded proteins were simultaneously upregulated, and 10 genes and their encoded proteins were simultaneously downregulated. Among them, *FD3* and *SFH10* participated in the pathway of ferredoxin (K02639), *GSTF11* participated in the pathway of glutathione S-transferase (K00799), *G6PD3* participated in the pathway of glucose-6-phosphate 1-dehydrogenase (K00036), and *SDH1-2* participated in the pathway of the proteasome regulatory subunit (K03030) (Table 1).

#### 2.1.2. GO Association Analysis of Transcriptome and Proteome

To understand the effect of exogenous MT on the physiological and biochemical characteristics of hairy broccoli roots, all screened DEGs and DAPs were subjected to GO correlation analysis, and the results were shown in Figure 2. The category of biological processes was mainly related to “single organism processes”, “metabolic processes”, “responses to stimulus”, and “cellular processes”. The category of cellular components was mainly related to “organelle part”, “extracellular”, “region membrane” and “membrane part”. The molecular function mainly related to “binding” and “catalytic activity”.

#### 2.1.3. KEGG Enrichment Analysis of the Transcriptome and Proteome

To further explore the biological pathways, and DEGs and DAPs function, KEGG pathway enrichment analysis was performed. The enrichment analysis is shown in Figure 3. The KEGG enrichment results for genes with opposite protein and mRNA expression trends (Figure 3a) were mainly in “Glutathione metabolism”, “Phenylpropanoid biosynthesis”, and “Fatty acid metabolism”. The KEGG enrichment results for proteins with opposite trends of protein and mRNA expression (Figure 3b) were mainly in “Pentose phosphate pathway”, “Fatty acid metabolism”, and “Fatty acid elongation”. The KEGG enrichment results for genes with the same protein and mRNA expression trends (Figure 3c) were in mainly “Phenylpropanoid biosynthesis”, “Fatty acid metabolism”, and “Fatty acid degradation”. The KEGG enrichment results for proteins with the same protein and mRNA expression trends (Figure 3d) were mainly in “Pentose phosphate pathway”, “Fatty acid metabolism”, and “Glutathione metabolism”. 

#### 2.1.4. Effect of MT Treatment on Glutathione Metabolism in Hairy Broccoli Roots

According to the KEGG enrichment results of genes with the same protein and mRNA expression trends (Figure 3c) and the KEGG enrichment results of proteins with the same protein and mRNA expression trends (Figure 3d), both proteins and mRNAs with the same expression trends were enriched to “Citrate cycle (TCA cycle) (ko00020)”, “Glutathione metabolism (ko00480)”, “Fatty acid degradation (ko00071)”, “Pentose phosphate pathway (ko00030)”, “Biotin metabolism (ko 00780)”, and “Fatty acid metabolism (ko01212)”. This study focused on the ‘Glutathione metabolism (ko00480)’ pathway, and showed differential expression of DEGs and DAPs of Glutathione S-transferases (GSTs) after treatment of exogenous MT in hairy roots. As shown in Figure 4, *GSTU9*, *GSTU22*, and *GSTF9* were upregulated at both the transcriptional and protein levels after MT treatment in hairy roots. *GSTL3*, *GSTU5*, *GSTU16*, *GSTU23*, *GSTU24*, and *GSTF10* showed differential expression at both transcriptional and protein levels after MT treatment in hairy roots.

### 2.2. Effect of MT Treatment on Physiological Indicators of Hairy Broccoli Roots

Exogenous MT causes an increase in the endogenous MT content of hairy roots. The effects of treating hairy broccoli roots with 500 μmol·L^−1^ MT on endogenous GSH content, and the GR activity of hairy roots in this study are shown in Figure 5. There was a highly significant increase in endogenous GSH content (Figure 5a), which was 38.00% higher than the control. There was a highly significant increase in endogenous GR activity (Figure 5b), which was 81.07% higher than the control. The above results indicated that MT treatment of hairy roots increased the endogenous GSH contents and GR activity in hairy roots. 

### 2.3. Effect of MT Treatment on Key Genes and Proteins Related to Glucoraphanin and Sulforaphane Synthesis in Hairy Broccoli Roots

This study analyzed the results of expression and protein quantification of key genes of MT-treated hairy broccoli roots. *CYP83B1*, *GSTF9*, *GSTF10*, *GSTU20*, *SUR1*, *UGT74B1*, *SOT18*, *SOT16*, and *FMOGS-OX5* regulate the synthesis of GRA, while *AOP1.2.2*, *GSL-OH1*, and *TGG4* regulate the synthesis of SF. The detection of upregulated gene expression and increased protein content of *CYP83B1* and *SOT16*-regulated genes indicated that MT treatment of hairy roots could promote the expression of *CYP83B1* and *SOT16* and the content of their regulated proteins. The relative expression of *GSTU20*, *SUR1*, *SOT18*, *AOP1.2*, *GSL-OH1*, and *TGG4* genes was decreased but the protein content was increased, and the analysis may be due to the presence of other regulations or modifications of these genes during translating proteins (Figure 6).

### 2.4. Validation of Transcriptome Data by qRT-qPCR Analyses

We verified the expression of GRA and SF synthesis-related genes in RNA-seq by qRT-PCR (Figure 7). The qRT-PCR results were consistent with the transcriptome data, indicating the reliability of transcriptomic data.

### 2.5. Effects of Exogenous GSH on the Synthesis of Glucoraphanin and Sulforaphane in Hairy Broccoli Roots

Based on transcriptomic and proteomic analysis results, the experiment investigated the effects of exogenously adding 100, 200, 300, and 400 μmol·L^−1^ of GSH to the hairy roots of broccoli for 12 h, on the synthesis of GRA and SF. The results, as shown in Figure 8, indicated that the addition of 400 μmol·L^−1^ GSH significantly increased the production of GRA in hairy roots compared to CK, with a 31.36% increase in GRA production compared to CK. The amount of GRA in the culture medium after GSH treatment was significantly higher than in the CK (Figure 8a). Furthermore, the production of SF in hairy roots was significantly higher in the 200, 300, and 400 μmol·L^−1^ GSH treatment group compared to the CK, with a 96.93% increase in SF production compared to CK. The amount of SF in the culture medium after 400 μmol·L^−1^ GSH treatment was significantly higher than in the CK (Figure 8b). In conclusion, the exogenous GSH can promote the synthesis of both GRA and SF in hairy roots.

### 2.6. Effects of Exogenous GSH on the Synthesis of Glucoraphanin and Sulforaphane as Well as the Expression of Glutathione-Related Genes in Hairy Broccoli Roots

Exogenous GSH treatment upregulated the expression of key genes involved in the synthesis of hairy root growth regulators GRA and SF, as shown in Figure 9. The expression of *SUR1* and *GLS-OH1* was downregulated. Additionally, GSH metabolism-related genes *GSTU5*, *GSTL3*, *GSTU9*, and *GSTU22* were upregulated. Therefore, exogenous GSH treatment promotes the synthesis of GRA and SF as well as GSH metabolism in hairy roots.

## 3. Discussion

This study added MT to the liquid suspension culture system of hairy broccoli roots, resulting in an MT concentration of 500 μmol·L^−1^ in the hairy root culture system. The complex molecular mechanism of MT-mediated GSH regulation of GRA and SF synthesis was revealed using transcriptomic and proteomic techniques combined with physiological indicators. A total of 5785 DEGs and 200 DAPs were found in hairy broccoli roots treated with 500 μmol·L^−1^ MT for 12 h. DEGs and DAPs GO association analysis was performed. As shown in Figure 2a–d, in the biological process category, it was mainly manifested in “single biological process”, “metabolic process”, and “response to stimuli and cellular process”. In the category of cellular components, it was mainly manifested in the categories of “organelle part”, “extracellular”, “region membrane”, and “membrane part”. In the molecular function category, it was mainly related to “binding” and “catalytic activity”. The KEGG enrichment analyses are shown in Figure 3a–d. Figure 3 revealed the presence of a “glutathione metabolism (ko00480) pathway” and further analysis revealed that DEGs and DAPs regulating “glutathione metabolism (ko00480)” were mostly downregulated (Figure 3a,b), while the “proteasome (ko 03050)” was downregulated by DEGs and DAPs (Figure 3c,d). MT alleviated drought stress in maize, and it was found that when 100 μmol·L^−1^ MT was sprayed on maize under drought stress, the MT-regulated genes were mainly associated with GSH metabolism [34], which was similar to the results of the present study, indicating that the metabolic pathways of plant GSH could respond to exogenous MT. 

According to the transcriptomic and proteomic data, this study found that exogenous MT increased the GSH content and GR activity in hairy roots (Figure 5). It has been shown that using melatonin to alleviate cadmium stress in maize increased GSH content [35], while the use of melatonin alleviated Cd stress and enhanced GR activity [36]. Wang et al. (2012) found that melatonin increased the concentration of GSH by stimulating the activity of γ-glutamylcysteine synthetase, thereby delaying the senescence of apple leaves [37]. This is consistent with the findings of this study, further demonstrating that exogenous MT can promote GSH biosynthesis.

We further examined the synthesis of secondary metabolites GRA and SF in hairy broccoli roots. Previously, our research group studies showed that exogenous MT significantly increased GRA and SF yields in hairy roots [15,16]. The differences in the expression of genes related to the synthesis of GRA and SF were analyzed by qRT-PCR under MT and GSH treatment in hairy roots. The results showed that both exogenous MT and GSH could upregulate the expression of *CYP83B1*, *UGT74B1*, *FMOGS-OX5*, *GSTF10*, and *GSTF11*, genes related to the synthesis of GRA and SF, and downregulate the expression of *SUR1* and *GLS-OH1*, genes also related to the synthesis of GRA and SF. However, exogenous MT downregulated the expression of *GSTF9*, while exogenous GSH upregulated the expression of *GSTF9* (Figure 7 and Figure 9). The possible reason for the difference is that the expression pattern of *GSTF9* is complex in response to MT and GSH, indicating that there are differences in the regulation of *GSTF9* expression by these two substances. GLS synthesis is a complex process and GSTs are a family of multifunctional enzymes involved in the biosynthesis of GLS. Some GSTs have been reported to be associated with secondary plant metabolism, particularly in the formation of natural products containing carbon-sulfur bonds, including sulfur-containing phytochemicals from cruciferous species [32,38]. GSH is ubiquitous and abundant in plants, and is distributed in mitochondria, plastids, nucleus, and cytoplasm, with the highest content in mitochondria, followed by nucleus, cytoplasmic matrix, and peroxisomes [39]. It plays an important role in the storage and transport of reduced sulfur and in the synthesis of proteins and nucleic acids, and it is also an effective regulator of enzyme activity [40]. In addition, GSH plays a key role in the maintenance of antioxidant properties of tissues and in the regulation of redox-sensitive signaling [41,42,43,44,45]. GSH provides sulfur atoms for *Arabidopsis* GLS synthesis [46]. During the biosynthesis of aliphatic GLS core structure, GSTs catalyze the formation of S-alkyl-thiohydroximate esters by conjugating the sulfur donor GSH with acyl nitrate compounds [32,33]. GSTs are encoded by multiple genes with *GSTF11*, *GSTU20*, *GSTF9*, and *GSTF10* participating in GLS synthesis [47,48]. In this study, exogenous MT was found to enhance the endogenous GSH content in hairy broccoli roots. Transcriptomic and proteomic analyses, as well as qRT-PCR detection, revealed the upregulation of *GSTU9*, *GSTU22*, and *GSTF11* expression in hairy roots treated with MT. Glutathione S-transferases (GSTs), particularly *GSTF11*, play a key role in the synthesis of glucosinolate-derived secondary metabolites, and are known to facilitate the conjugation of substrates with GSH. In addition, the exogenous MT upregulated the expression of *GSTU9* and *GSTU22*, promoting the transport of sulfur donors and further inducing the synthesis of GRA. Studies have shown that the deterioration of broccoli quality after harvest is due to the downregulation of the expression of *GSTF11*, *GSTU20*, and *ST5b* genes encoding sulfur donor transport proteins during the storage process, which hinders the transport of sulfur donors and therefore blocks the synthesis of GRA [49]. This study further confirmed the findings of our research. This further explains the promotion of GLS biosynthesis by exogenous MT in hairy roots. The study also demonstrated that exogenous GSH supplementation stimulated the synthesis of glucosinolate-derived GRA and SF, as depicted in Figure 8. qRT-PCR analysis revealed the upregulation of key genes associated with GLS and GSH metabolism, namely *GSTU9*, *GSTU22*, and *GSTF11*, in hairy broccoli roots treated with GSH. Collectively, these findings indicate that the exogenous addition of MT promotes endogenous GSH synthesis and that both exogenous MT and GSH enhance the upregulation of the expression of *GSTF11* genes encoding sulfur donor transport proteins, thereby promoting the synthesis of GRA in hairy roots. Based on the experimental results, a diagram illustrating the MT-mediated GSH regulation of glucosinolate GRA and SF in hairy broccoli roots was constructed (Figure 10).

## 4. Materials and Methods

### 4.1. Plant Materials

The hairy broccoli root culture system established in our laboratory [15] was used to induce the production of hairy roots in the leaves of sterile broccoli seedlings of “Zhongqing 9” using *Agrobacterium rhizogenes* ATCC15834. In an ultra-clean bench (Suzhou Jedi Purification Technology Co., Ltd., Suzhou, China), 1 mg·mL^−1^ of hairy roots were inoculated in 100 mL of MS liquid medium and incubated in a bed temperature incubator. The incubation conditions were 25 °C and 110 rpm·min^−1^ speed in the dark.

### 4.2. Experimental Design

Based on the results of our previous laboratory study, MT was added to the broccoli hairy root culture system within 20 days to achieve a final concentration of 500 µmol·L^−1^ of MT in the culture system, and the material was harvested after being treated for 12 h. Untreated hairy roots were used as the control, and a portion of the treatment material was sent to Biomarker Technologie (Beijing, China) for transcriptome determination and quantitative proteome analysis using Illumina sequencing technology. The genome of wild cabbage (version: GCA_900416815.2) was used as the reference genome for transcriptome determination and quantitative proteome analysis using TMT technology. Another part of the material was extracted and tested for endogenous GSH contents and root activity of hairy roots in the hairy root culture system. All experiments were performed with three biological replications.

GSH was added to the hairy broccoli root culture system within 20 days to achieve final concentrations of GSH at 100, 200, 300, and 400 µmol·L^−1^ in the culture system, and the material was harvested after being treated for 12 h. Untreated hairy roots were used as the control to detect the production of GRA and SF in the hairy root culture system and select the optimal concentration of GSH treatment. Finally, 400 µmol·L^−1^ of GSH was selected to treat hairy roots for qRT-PCR detection of GRA and SF synthesis, as well as the expression of genes related to GSH metabolism.

### 4.3. Experimental Methods

#### 4.3.1. Solution Configuration

Preparation of 20 mmol·L^−1^ MT solution: 0.2330 g of MT was precisely weighed, and dissolved with anhydrous ethanol in an ultra-clean bench (Suzhou Jedi Purification Technology Co., Ltd., Suzhou, Jiangsu, China). The solution was slowly fixed with sterile water to 50 mL, and filtered in a sterilized centrifuge tube with a 0.22 μm sterilized filter tip (Shanghai Anpu Experimental Technology Co., Ltd., Shanghai, China). Light was avoided throughout the operation. It was stored at 4 °C in a refrigerator to avoid light, and re-dissolved in a water bath with heat before use if precipitate was produced. MT was purchased from Shanghai Yuanye Biotechnology Co., Ltd. (Shanghai, China) with a purity of 99% and CAS number: 73-31-4.

Preparation of 20 mmol·L^−1^ GSH solution: 0.3073 g of GSH was precisely weighed. The solution was slowly fixed with sterile water to 50 mL, and filtered in a sterilized centrifuge tube with a 0.22 μm sterilized filter tip (Shanghai Anpu Experimental Technology Co., Ltd., Shanghai, China). It was stored at 4 °C in a refrigerator, and re-dissolved in a water bath with heat before use. GSH was purchased from Shanghai Yuanye Biotechnology Co., Ltd. in China with a purity of 99% and CAS number: 70-18-8.

#### 4.3.2. Extraction and Detection of Reduced Glutathione in Hairy Roots

Referring to the method of Chen et al. (2002) with slight modifications [50], 0.5 g of hairy roots was added to 5 mL of 5% trichloroacetic acid and then ground, and centrifuged at 15,000× *g* for 10 min. The supernatant was fixed to 5 mL. The above sample extracts were taken; 0.25 mL, and 150 mmol·L^−1^ NaHPO_4_ (pH 7.7) 2.6 mL and DTNB (2-nitrobenzoic acid) 0.15 mL were added to each, and a phosphate buffer was added instead of DTNB as a blank control. After shaking well, the reaction was held at 30 °C for 5 min, and the absorbance value at 412 nm was measured. The GSH content of the samples was calculated according to the standard curve.

#### 4.3.3. Extraction and Detection of Glutathione Reductase Activity in Hairy Roots

Referring to the method of Halliwell et al. (1978) [51], a small amount of quartz sand was added along with 3 mL of pre-chilled extract liquid (0.1 mol·L^−1^ KH_2_PO_4_-KOH buffer, pH 7.5, containing EDTA (1 mg·mL^−1^)). It was ground thoroughly in a mortar and transferred to a clean centrifuge tube. Then, the mortar and pestle were washed with 2 mL of extract liquid. The combined reaction mixture was 50 mmol·L^−1^ Tris-HCl buffer (containing 0.1 mmol·L^−1^ EDTA, 5 mmol·L^−1^ MgCl_2_, pH 7.5). For the assay, the reaction mix, 10 mmol·L^−1^ reduced nicotinamide adenine dinucleotide phosphate (NADPH) and 10 mmol·L^−1^ oxidized glutathione (GSSG), was pre-warmed in a 25 °C water bath. To 780 μL of reaction mix, 150 μL of enzyme solution, 0.2 mmol·L^−1^ NADPH, and 0.5 mmol·L^−1^ GSSG were added to initiate the reaction at a final concentration of 1 mL. Reaction values were read every 30 s and the decrease in OD340 was read from 0.5 to 3.5 min to calculate enzyme activity.

### 4.4. RNA-Seq Library Construction and Illumina Sequencing

The transcriptome data in this study are the same as the Tian et al. (2021) data with a different study focus [15].

Sample detection, library construction, library quality control, computer sequencing, and sequence alignment with the reference genome wild cabbage (*Brassica oleracea* L. var. *italica* Planch) was used as the reference genome (version: GCA_900416815.2) for sequence alignment and differential expression analysis.

#### 4.4.1. RNA Extraction Library Construction and Sequencing

Hairy roots were treated with MT for 0 and 12 h. The MT-treated hairy roots and control samples (0 h) were collected from three biological replicates and analyzed by transcriptomic-based technology. RNA concentration and purity were measured using a NanoDrop 2000 (Thermo Fisher Scientific, Wilmington, DE, USA). RNA integrity was assessed using the Illumina HiseqTM 2500 by Biomarker Technologies Co, Ltd., (Beijing, China).

#### 4.4.2. Differentially Expressed Gene Analysis

To identify differentially expressed genes (DEGs) across samples or groups, the edgeR package (http://www.r-project.org/, accessed on 15 June 2022) was used. The results identified genes with a fold change ≥2 and a false discovery rate (FDR) < 0.05 in a comparison as significant DEGs. DEGs were then subjected to enrichment analysis by GO (http://www.geneontology.org/, accessed on 15 June 2022) functions and KEGG pathways. KEGG was the major public pathway-related database. 

### 4.5. Proteomics

#### 4.5.1. Protein Extraction, Protein Digestion, and Tandem Mass Tags Labeling

Proteins were extracted from hairy broccoli roots using trichloroacetic acid/acetone. Protein extraction and digestion were performed as described previously [52]. Briefly, the total protein extracted from each sample (100 μg) was mixed with 100 μL of lysis buffer. This was added with tris(2-chloroethyl) phosphate (10 mmol·L^−1^) and stored at 37 °C. After 1 h, iodoacetamide (40 nmol·L^−1^) was added to the mixture and stored at room temperature and protected from light for 40 minutes. Then, six times the volume of cold acetone was added to precipitate the protein at −20 °C for 4 h. The mixture was centrifuged at 10,000× *g* for 20 min at 4 °C and the precipitate was resuspended in 100 µL of 50 mmol·L^−1^ TEAB buffer. Trypsin was added to the protein at a ratio of 1:50 and the mixture was incubated overnight at 37 °C. One unit of Tandem Mass Tags reagent was thawed and reconstituted in 50 µL acetonitrile at room temperature for 2 h. Hydroxylamine was added and reacted at room temperature for 15 min. Finally, all samples were mixed, desalted, and vacuum dried. All experiments were performed with three biological replications.

#### 4.5.2. LC–MS/MS Analysis

The 9RKFSG2_NCS-3500R system (Thermo, Wilmington, DE, USA) was connected to the Q_Exactive HF-X system (Thermo, Wilmington, DE, USA) via an electro-spray ionization for the study. Analysis of labeled peptides by on-line nanoflow liquid chromatography-tandem mass spectrometry. Briefly, a C18 reversed-phase column (75 μm × 25 cm, Thermo, Wilmington, DE, USA) was equilibrated with solvent A (2% formic acid and 0.1% formic acid) and solvent B (80% acetonitrile and 0.1% formic acid). The elution conditions were as follows: 0–2 min, 0–3% B gradient elution; 2–92 min, 5–25% B; 92–102 min, 25–45% B; 102–105 min, 45–100% B; 105–120 min, 100–0% B; flow rate, 300 nL·min^−1^. The Q_Exactive HF-X operates in Data Dependent Acquisition (DDA) mode and can automatically switch between full scan MS and MS/MS acquisition. In Orbitrap, a full scan mass spectrum was obtained in the *m*/*z* 350–1500 range with a resolution of 70,000. The automatic gain control (AGC) target was 3 × 10^6^ and the maximum fill time was 20 ms. The first 20 parent ions were selected to enter the collision cell for high-energy collisional dissociation (HCD) fragmentation. MS/MS resolution was set to 35,000 (*m*/*z* 100); automatic gain control (AGC) target was set to 1 × 10^5^; maximum fill time was 50 ms and dynamic rejection time was 30 s. All experiments were performed with three biological replications.

#### 4.5.3. Protein Identification and Data Analysis

Raw data were analyzed using Proteome Discoverer (Thermo Scientific, version 2.2). MS/MS search conditions were as follows: mass tolerance 20 ppm Da 0.02 MS and MS/MS tolerance; trypsin 2 missed cleavage was allowed. Ureido-cysteine methylation and TMT *N*-terminus and lysine side chain peptide were used as fixed modifications, and methionine oxidation was used as dynamic modifications, respectively. The false discovery rate for peptide identification was set to FDR ≤ 0.01. At least one unique peptide identification was used to support protein identification. GO (https://geneontology.org/, accessed on 18 June 2022) and KEGG pathways (https://www.genome.jp/kegg/, accessed on 18 June 2022) were annotated for all identified proteins.

#### 4.5.4. Data Processing

The database was selected with reference to the desired species, completeness of database annotation, and sequence reliability.

### 4.6. Real-Time Quantitative PCR Analysis

Twelve DEGs-specific primers were designed by primer 5 (Table 2). Total RNA was isolated using an RNA kit (Tiangen DP432) (Tiangen Biochemical Technology Co., Ltd, Beijing, China). cDNA synthesis and qRT-PCR analysis were performed using a one-step SYBR Prime script plus an RT-PCR kit (Tiangen FP209) (Tiangen Biochemical Technology Co., Ltd., Beijing, China). According to the following procedure, PCR amplification was performed in a 96-well platform (Roche LC 96) (Roche, Basel, Switzerland): 180 s at 95 °C, 5 s at 95 °C, 15 s at 60 °C, and 40 cycles. The melting curve analysis was performed at 60~95 °C. According to the Ct value of the target gene and the internal reference gene, the gene expression level was calculated by the 2^−ΔΔCt^ method [53]. The *ADF3* gene was selected to be an internal control.

### 4.7. Data Analysis

In this study, each group was independently replicated three times, and the experimental results were expressed as the mean ± standard error of the three biological replicates; ANOVA with SPSS v. 21.0 was used for analysis of variance, and the Duncan’s new complex polarization method was used to test the significance of differences (*p* ≤ 0.01). Data statistics were analyzed in Excel v. 2021, and column charts were performed by origin v. 2018. Heatmaps were drawn by using TBtools v. 1.0 normalized function to normalize the data.

## 5. Conclusions

In this study, the gene expression pattern and protein content of MT-treated hairy broccoli roots were analyzed by the combination of RNA-seq and TMT techniques. KEGG analysis revealed that both proteins and mRNAs with the same expression trend were mainly enriched in “Glutathione metabolism (ko00480)” and “Proteasome (ko03050)”. “Proteasome (ko03050)”, and most of the DEGs and DAPs regulating these two pathways were downregulated. The results showed that endogenous GSH concentration and GR activity in hairy roots increased after MT treatment. Exogenous addition of GSH could promote the biosynthesis of GRA and SF, and both exogenous MT and GSH could upregulate the expression of sulfur transport-related gene *GSTF11*, thus promoting the biosynthesis of GRA.

## Figures and Tables

**Figure 1 plants-13-00106-f001:**
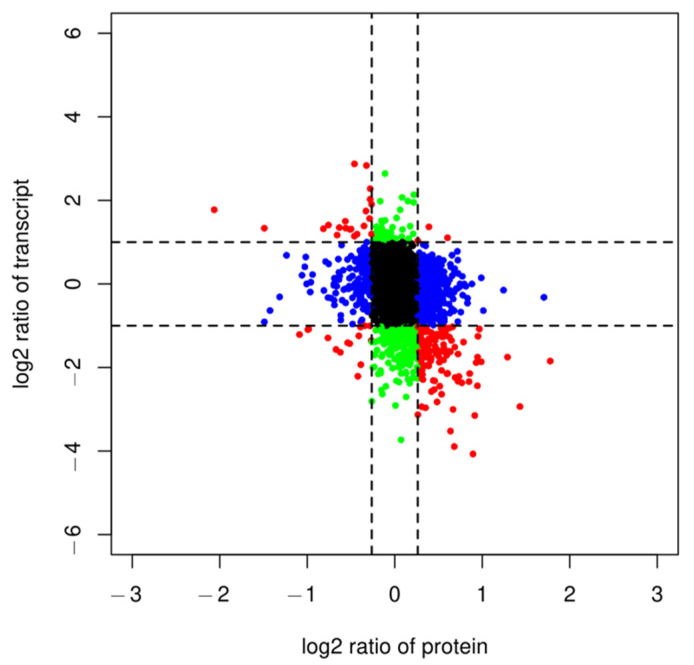
Nine-quadrant diagram of protein and transcriptome expression patterns in different groups. Quadrants 1 and 9 indicate opposite protein and mRNA expression trends (red dots); quadrants 3 and 7 indicate the same protein and mRNA expression trends (red dots); quadrants 2 and 8 indicate no differential expression in protein and differential mRNA expression (green dots); quadrants 4 and 6 indicate no differential expression in mRNA and differential protein expression (blue dots); and quadrant 5 indicates no differential expression of both protein and mRNA (black dots).

**Figure 2 plants-13-00106-f002:**
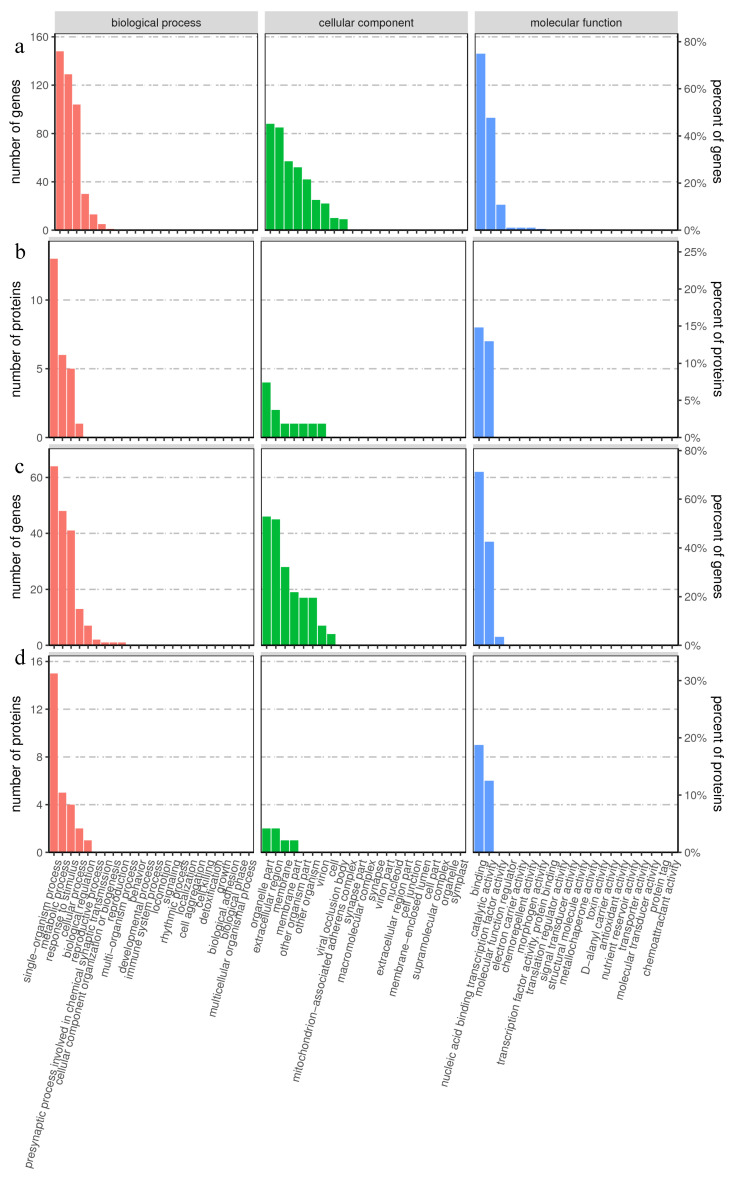
GO association analysis of proteomic and transcriptomic data. (**a**) GO classification maps of genes with opposite trends in protein and mRNA expression. (**b**) GO classification maps of proteins with opposite trends in protein and mRNA expression. (**c**) GO classification maps of genes with the same protein and mRNA expression trends. (**d**) GO classification maps of proteins with the same protein and mRNA expression trends.

**Figure 3 plants-13-00106-f003:**
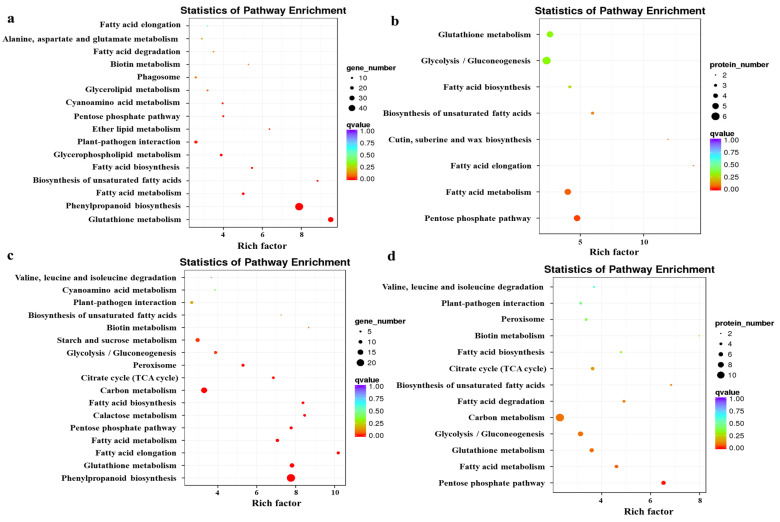
Enrichment analysis of transcriptomic and proteomic data by KEGG. (**a**) KEGG enrichment results for genes with opposite protein and mRNA expression trends. (**b**) KEGG enrichment results for proteins with opposite trends in protein and mRNA expression. (**c**) KEGG enrichment results for genes with the same protein and mRNA expression trends. (**d**) KEGG enrichment results for proteins with the same protein and mRNA expression trends. The enrichment factor indicates the ratio of genes and proteins annotated to differential gene and protein pathways to all genes annotated to the pathway. The enrichment factor indicates that the enrichment level of differential genes in this pathway was more significant. The *p*-value corrected for multiple hypothesis testing is the *q*-value, with smaller *q*-values representing more reliable significant enrichment.

**Figure 4 plants-13-00106-f004:**
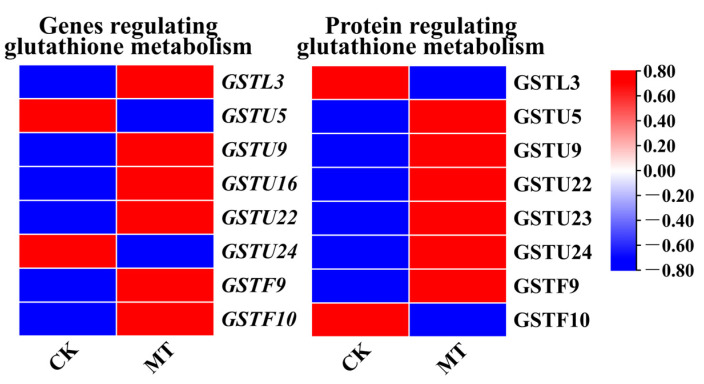
Expression of DEGs and DAPs involved in glutathione metabolism in MT-treated hairy broccoli roots. Three independent repeats.

**Figure 5 plants-13-00106-f005:**
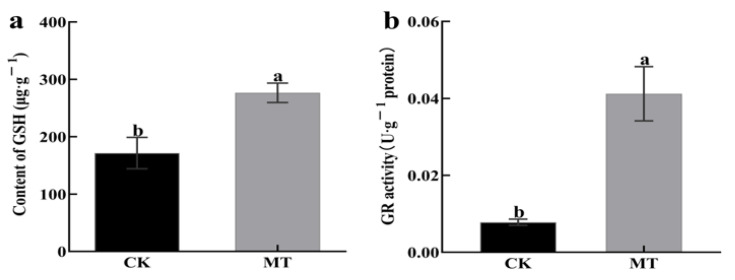
Effect of adding exogenous MT on the physiological indexes of hairy broccoli roots. (**a**) Effect of adding exogenous MT on endogenous GSH content of hairy broccoli roots. (**b**) Effect of adding exogenous MT on the GR activity of hairy broccoli roots. Data exhibit means ± SE of three replicates. Different small letters in the figure showed a significant difference (*p* ≤ 0.05).

**Figure 6 plants-13-00106-f006:**
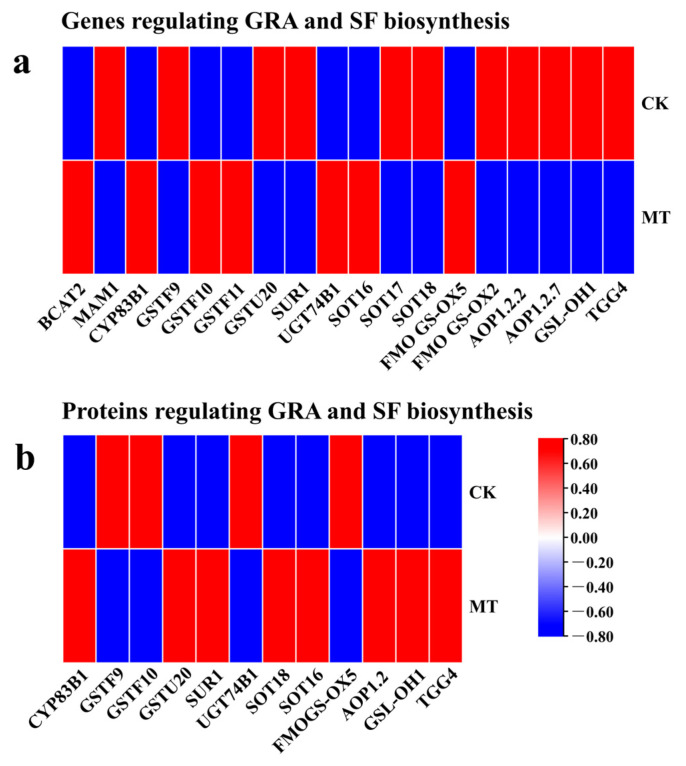
The relative expression of key genes expressed and the relative content of key proteins in MT-treated hairy broccoli roots. (**a**) The relative expression of key genes in MT-treated hairy broccoli roots. (**b**) The relative content of key proteins in MT-treated hairy broccoli roots. Three independent repeats.

**Figure 7 plants-13-00106-f007:**
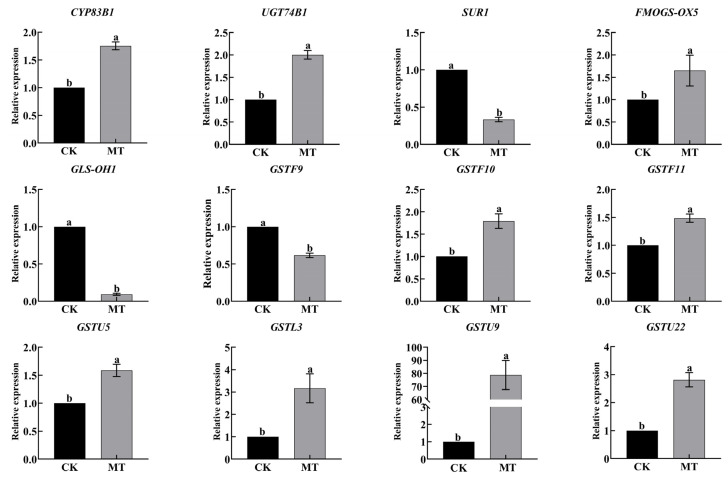
Expression of GRA and SF synthesis and glutathione metabolism-related genes by qRT-PCR. Data exhibit means ± SE of three replicates. Different small letters in the figure showed a significant difference (*p* ≤ 0.05).

**Figure 8 plants-13-00106-f008:**
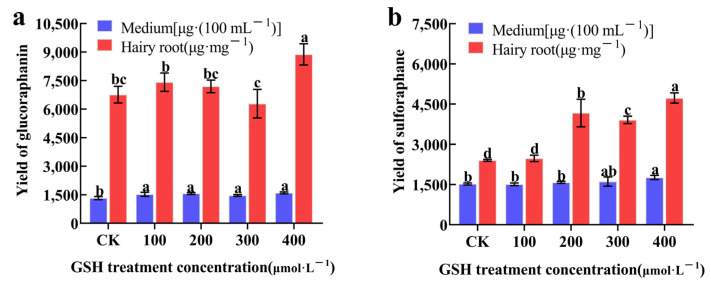
The effects of exogenous GSH on the synthesis of GRA (**a**) and SF (**b**) in hairy broccoli roots. Data exhibit means ± SE of three replicates. Different small letters in the figure showed a significant difference (*p* ≤ 0.05).

**Figure 9 plants-13-00106-f009:**
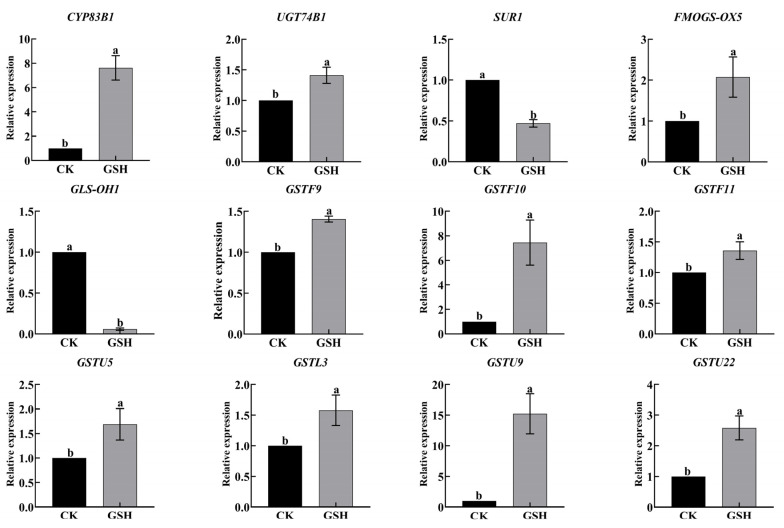
Effects of exogenous addition of GSH on the synthesis of glucoraphanin and sulforaphane, as well as the expression of glutathione-related genes in hairy broccoli roots. Data exhibit means ± SE of three replicates. Different small letters in the figure showed a significant difference (*p* ≤ 0.05).

**Figure 10 plants-13-00106-f010:**
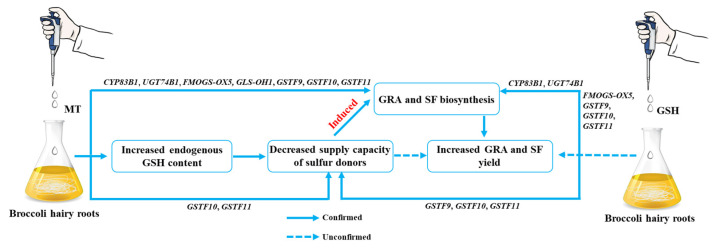
MT and GSH regulate GRA and SF synthesis.

**Table 1 plants-13-00106-t001:** Common differentially expressed genes between the proteome and the transcriptome.

Protein Name	Regulation	Gene Name	Regulation	GO Annotation	KEGG Annotation
FD3	↑	*FD3*	↑	--	K02369
SG1	↑	*SG1*	↑	Molecular Function	--
GSTF11	↑	*GSTF11*	↑	Molecular Function	K00799
MIK2	↑	*MIK2*	↑	Cellular Component	--
A2	↑	*A2*	↑	Cellular Component	--
UCC2	↑	*UCC2*	↑	Molecular Function	--
CCP2	↑	*CCP2*	↑	Molecular Function	K12820
CYTB5-D	↑	*CYTB5-D*	↑	Molecular Function	--
SGS3	↑	*SGS3*	↑	Molecular Function	--
G6PD3	↑	*G6PD3*	↑	Cellular Component	K00036
SFH10	↑	*SFH10*	↑	Molecular Function	K02639
PG11	↓	*PG11*	↓	Biological Process	K01904
KIN14I	↓	*KIN14I*	↓	Molecular Function	K01810
IAR4	↓	*IAR4*	↓	Molecular Function	K10406
FUM1	↓	*FUM1*	↓	Molecular Function	K00161
NUDT7	↓	*NUDT7*	↓	Molecular Function	K01679
RPN11	↓	*RPN11*	↓	Molecular Function	--
SDH1-2	↓	*SDH1-2*	↓	Cellular Component	K03030
GAL1	↓	*GAL1*	↓	Cellular Component	K00234
PED1	↓	*PED1*	↓	Molecular Function	K18674
4CL4	↓	*4CL4*	↓	Molecular Function	K07513

**Table 2 plants-13-00106-t002:** Primer sequences used in qRT-PCR analysis.

Gene Name	Primer Sequences (5′ to 3′)	Accession
*ADF3*	Forward: GAGGAGCAGCAGAAGCAAGTGG	XM_013797493.2
	Reverse: ATCGGCATTCATCAGCAGGAAGAC	
*CYP83B1*	Forward: TTTCGGGTCAGGCAGAAGAATGTG	XM_013749594.1
	Reverse: ATCCCTGTCGGTAGGCTCCAATC	
*UGT74B1*	Forward: ACAACAGCGACCAACTCCAAAGG	XM_013730524.1
	Reverse: GTGTAGGTGGTGGTGGCGATTG	
*SUR1*	Forward: CGAAGAACAAGCACACGCCAAC	XM_013743483.1
	Reverse: CTCCACCGAACCGCCAAACTG	
*GSL-OH*	Forward: ACGAGATGAAAATCGGCGTGAAGG	XM_013781535.1
	Reverse: TGGCTTGCGGGTTATGGAATATGC	
*FMOGS-OX5*	Forward: TGGCAGTGATCGGAGCAGGAG	XM_013749606.1
	Reverse: GAAGACGACGACGGAGTGTGATTC	
*GSTF9*	Forward: GCTCTGGTAACGCTCATCGAGAAG	XM_013827777.3
	Reverse: AAGGCGAGATAAGCAGGCTGTTTG	
*GSTF10*	Forward: CCGATTTGGCTCACCTTCCCTTC	XM_013766931.1
	Reverse: TTCCAGGCAGCACGGTTACTAATC	
*GSTF11*	Forward: ATCTTCTTCGTCAGCCGTTTGGTC	XM_013730029.1
	Reverse: CCGTTCCTTGGTCCGCATACTTG	
*GSTU5*	Forward: TTTGTATCAATGGCAAGAGCAGACG	XM_013889167.3
	Reverse: TCCGACAAGTTCCTTCTCCAGATTC	
*GSTL3*	Forward: CCGATCCACCCGCTCTGTTC	XM_013760577.1
	Reverse: GCAGGCACCTTGTTTTCAGGATAG	
*GSTU9*	Forward: CTGACTAACGAGACTATGAGCCTTG	XM_013756220.1
	Reverse: GCCAACTACAGACACCAGGAAC	
*GSTU22*	Forward: ATGGCTGATGAGGTGATTCTTCTAG	XM_013765269.1
	Reverse: TCAGTGCGATCCTTGCTCTTAC	

## Data Availability

The sequencing raw data have been uploaded to the Sequence Read Archive (https://www.ncbi.nlm.nih.gov/sra, accessed on 18 June 2022) under Bioproject PRJNA764437. The reference genome and gene model annotation files of Broccoli (*Brassica oleracea* L. var. *botrytis* L Planch) were downloaded from genome websites (GCA_900416815.2).

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
