# Peer review of "Exogenous Melatonin Promotes Glucoraphanin Biosynthesis by Mediating Glutathione in Hairy Roots of Broccoli (Brassica oleracea L. var. italica Planch)"

_plants, 2023, doi:10.3390/plants13010106_

Round 1

Reviewer 1 Report

Comments and Suggestions for Authors

Previously, MT mediated GSH regulation of GRA and SF biosynthesis had not been reported. Therefore, this work is very interesting. They verified the expression of GRA and SF synthesis-related genes. However, the manuscript lacks some important information. Please specify how analysis of GRA and SF contents was performed. Please present representative chromatograms of the compounds from samples.

minor issues

Line 14: Abbreviations need to be defined at first mention.

The resolution of Fig. 3 is very low.

Line 223: Change “The” to “the”.

Figure 4-7: Please specify CK and MT. Add repeat number (n = ?) in figure caption.

Figure 8: Add repeat number (n = ?) in figure caption.

Figure 9: Is GSH MT?

Comments on the Quality of English Language

The quality of English language is good.

Author Response

Dear reviewer

Thank you very much for your careful and thoughtful comments and suggestions on the manuscript titled “Exogenous melatonin promotes glucoraphanin biosynthesis by mediating glutathione in hairy roots of broccoli (Brassica oleracea L. var. italica Planch)”, We have carefully considered your comments. In addition, we had the manuscript edited by the school's native English-speaking international student editors to ensure proper English usage, grammar, punctuation, and spelling. Revisions in the manuscript are marked in red. Our response to your comments is shown below.

Best wishes

Li Sheng

Reviewer 2 Report

Comments and Suggestions for Authors

The authors investigated effects of melatonin (MT) treatment in hairy roots of broccoli in terms of transcriptomic and proteomic changes. Correlation analysis could extract differentially expressed genes that were commonly changed between transcript and protein profiles (Table 1). Due to lacking of detailed descriptions about the genes in Table 1, I could not interpret what the authors wanted to say by Table 1. For example, annotation of K00036 is “glucose-6-phosphate 1-dehydrogenase [EC:1.1.1.49 1.1.1.363]” belonging to pentose phosphate pathway. But the authors did not mention about the gene later. Furthermore, I could not understand what BolC3t15525H.gene is. Protein and gene IDs are same and no description about gene annotation. The authors had an interest of genes and proteins that were same/opposite expression/accumulation trends. But the authors just focused on genes/proteins that showed similar expression trends. Why? No descriptions about it. Omics analysis should be objectively investigated. In section 2.1.4, the authors showed the changes of genes and proteins involved in glutathione metabolism. There are many homologues in glutathione production and degradation as in Figure 4. However, the authors did not show how many GSTLs, GSTUs and GSTFs were detected in the transcript and protein profiles. Which genes/proteins (enzymes?) have major function for glutathione production? I could not find statistical evaluation of expression/accumulation of genes/proteins in Figure 4. Fold change |0.80| is too low. FDR or q-value should be shown.

In the previous study, the authors already investigated transcriptomic changes in the same materials triggered by MT treatment. By KEGG enrichment analysis, the authors focused on Glucosinolates biosynthesis, in particular GRA. Why did the authors show changes of transcript/protein profiles related to GRA? The authors may want to show SF biosynthesis through GRA degradation catalyzed by myrosinase. But there were no explanations what and how many genes/proteins are involved in SF biosynthesis (I could find information about GRA biosynthesis in Tian et al (2021).  

It would be great if the authors could show schematic model of (at least) gene regulations in GRA, SF and glutathione biosynthesis triggered by exogenous addition of MT and GSH. What gene expressions were different when compared to MT- and GSH treatments, respectively? Structures and biosynthesis of MT and GSH is completely different. Did GSH treatment could enhance endogenous MT production in hair roots of broccoli? I think the authors should make clear the point.

Comments on the Quality of English Language

The manuscript is well written, but there are tiny mistakes. For example, “N” should be italic in N-acetyl-5-methoxytryptamine. Please check the manuscript carefully.

Author Response

(The authors gave the same response as above.)

Round 2

Reviewer 2 Report

Comments and Suggestions for Authors

About Figure 4 and 5. The authors responded to my comments about heat map in authors’ response. But I could not understand (i) what is the control to calculate (log2?) fold changes, (ii) how the authors normalized data and (iii) what color bar was represented (log2-fold change??). It should be shown whether differential expression of genes and relative content of proteins were significantly changed or not supported by statistical evaluation.    

Author Response

Dear Reviewer

Thank you very much for your careful and thoughtful comments and suggestions on the manuscript titled "Exogenous melatonin promotes glucoraphanin biosynthesis by mediating glutathione in hairy roots of broccoli (Brassica oleracea L. var. italica Planch)". We have carefully responded to your comments. The revisions in the manuscript are marked in red. Our response to your comment is as follows.

Best wishes

Li Sheng

Round 3

Reviewer 2 Report

Comments and Suggestions for Authors

The authors explained how to generate heatmaps using TBtools v. 1.0. They added a sentence in the revised manuscript. I am satisfied with it.